# Dynamic Voltage Regulation and Unbalance Compensation in a Low-Voltage Distribution Network Using Energy Storage System

Krzysztof Rafał *, Jacek Biskupski, Sebastian Bykuć and Patryk Chaja

Institute of Fluid Flow Machinery, Polish Academy of Sciences, 80-231 Gdańsk, Poland
* Correspondence: krafal@imp.gda.pl

**Featured Application: Energy Storage Systems, Active Power Conditioning.**

**Abstract:** Modern distribution grids may suffer problems of voltage distortion, especially along radial low-voltage feeders with a high penetration of intermittent, unbalanced and distorted loads and generation sources. It is a challenge to develop an effective voltage-regulation method using a straightforward implementation. This paper proposes a novel method for local voltage control and balancing using a shunt-connected energy storage system. The compensation principles are explained, and a complete controller design is proposed. The algorithm is designed to be implemented in power electronic converters that provide the interface between the storage and the grid. The original contribution is the development of a low-level control method, which includes voltage balancing and a method to minimize the compensator current and is to be implemented in power electronic converters that provide the interface between the storage and the grid. The calculation of active and reactive compensator currents is explained with relation to the estimated grid impedance. The efficacy of the designed controller is verified by laboratory experiments. It is shown that voltage regulation using the proposed method is achieved with less apparent power compared to a system where only reactive power is used. The controller presents a very good dynamic response to rapid voltage variations, such as unbalanced voltage dips. The applicability and constraints of the method are discussed with respect to the present state of the art in low-voltage-grid voltage regulation.

**Keywords:** voltage control; power quality; energy storage





## 1. Introduction

In recent years, a significant transformation of distribution grids has been observed. Global policies to decrease fossil fuel consumption have, in many countries, led to regulations that favor clean energy technologies [1,2]. Many new solutions for producing and utilizing electrical energy become attractive alternatives for end customers. As a result, a growing number of photovoltaic (PV) installations, small wind turbines, electric heating systems (i.e., heat pumps and electric boilers) and electric vehicles (EV) have been observed [3]. These relatively heavy loads and sources are usually connected at the end customer level, directly to the low-voltage (LV) grid. This fact poses technical challenges for the existing grid infrastructure, which now has to deal with heavy loads and power generation of an intermittent nature.

Among others, one of the major challenges is maintaining the voltage level in the distribution grid within allowable limits as well in static and dynamic states. In addition, single phase loads may introduce a significant voltage unbalance, power converters may introduce harmonic currents and fluctuating loads may introduce voltage flicker. The negative impact of such situations may include, e.g., damage to or malfunction of the electrical equipment and limitations of the hosting capacity of the Distributed Energy Resources (DER) in the distribution grid [4,5].

Having many factors that influence the voltage profile [6], power quality issues are mostly observed in rural areas, typically characterized by radial topology and utilization of overhead lines with high feeder impedance.

Methods for voltage regulation in distribution networks may include [7,8]:

- Transformers with on-load tap changers (OLTC)—only at the substation point, but they provide no information or control of the voltage along the feeder;
- Network reinforcement—an increased cross section or meshed network decreases grid impedance and, thus, voltage drop;
- Custom power systems—a variety of costly power-electronics-based solutions dedicated to industry [9];
- DER power curtailment or load drop—though effective, causes a reduction in user comfort and/or revenue;
- Demand side management—requires a costly information and communication technologies (ICT) infrastructure;
- Reactive power control—found to be ineffective and difficult to coordinate in LV networks;
- Optimal integration of DER in the distribution system [10];
- Introduction of energy storage systems (ESS)—a costly solution; however, a decrease in prices is expected, making it a possible, attractive solution [1].

One of the most prospective techniques, which avoids extra investments, is the implementation of voltage-regulation strategies in the power electronic converters that provide the interface between DERs, EV chargers and ESS on one side and the grid on the other. The interaction of a DER with the distribution network is regulated by grid codes, which typically utilize generation units for voltage regulation in a very limited way. Even if technically feasible [11], grid codes do not refer to voltage unbalance, which may become a serious issue with the growing penetration of single-phase loads or single-phase generation. It is easy to violate voltage limits in separate phases, while three-phase values are within the limits. Grid codes for storage systems and interactive grid EV charging stations are being developed [12,13].

Local voltage regulation by reactive power injected by PV inverters [14] and by active power curtailment [15,16] are both extensively studied methods. The efficiency of both methods highly depends on grid-impedance parameters. The first method is easily adopted; however, it requires good coordination of the PV inverter set points and depends on the grid's short circuit parameters. In LV grids, especially in rural areas with overhead lines, high R/X ratios are observed. In this case, reactive power has little influence on voltage modules and creates additional losses [17]. Active power curtailment is a viable solution for this scenario; however, it results in a reduction in revenue for the DER owner.

Efficient power quality and overall operational improvement of LV grids can be achieved by introducing energy storage systems, due to their flexibility in active power control [18,19]. Studies show that voltage-dependent ESS control strategies can provide a good trade-off between power curtailment and voltage regulation [20]. Other authors propose coordinated PV and ESS control for effective voltage regulation, indicating differences between urban and rural scenarios [17]. Voltage regulation can also be achieved by deploying ESS at the distribution system operator (DSO) [21].

Some reported voltage-regulation methods propose centralized optimization and scheduling/dispatch [21–23]; others are based on local controllers [20,24–26]. For LV systems, local methods are preferred, as they do not require a costly ICT infrastructure.

In LV grids, due to the increasing share of sizeable single-phase loads, voltage unbalance also has to be addressed. Unbalance compensation may be achieved by reactive power control and optionally active power curtailment of single-phase inverters [27,28]. Additionally, unbalanced control of ESS, PV and EV charging is proposed for unbalanced conditions [22,24,26,29].

Typically in grid applications, droop control methods that allow sharing of regulation effort are applied, but these are not always efficient in distributed voltage control [16].

Dynamic voltage regulation requires control schemes similar to DSTATCOM, which utilize closed-loop voltage controllers [25,30].

A conventional voltage-control method would include one of the above solutions. The goal of the conducted research was to identify a complete and easy-to-implement method for the overall improvement of the voltage quality in LV networks. The authors have not found a complex solution that includes dynamic voltage regulation, balancing and harmonics minimization.

The novel solution proposed in this paper is an application of closed-loop voltage controllers based on local voltage measurement, with characteristics for DSATCOM devices and energy storage systems installed in LV grids. A shunt-compensation device is preferred over series compensation, as it is easier to install compared to series devices and allows for transformerless operation in LV grids. Additionally, the shunt compensator provides voltage profile improvement along the whole feeder, not only downstream of the compensating device.

The contribution of this paper is the development of a control structure for the power electronic converter, which provides the interface between an energy storage and the distribution network. The designed control structure ensures universal power quality enhancement by a dynamic control of voltage at the point of coupling, as well as voltage balancing by negative sequence current injection. The novelty of the research is a method of setting active and reactive compensator current components, with respect to grid impedance parameters, allowing to minimize the current used for voltage regulation compared to other solutions such as reactive power control [14,29] or active power dispatch [21,22].

The paper is organized as follows. Section 2 describes the principle of the shunt compensation and minimization of compensator current, the design of converter control for voltage regulation and balancing. Section 3 shows the results of the laboratory experiment, proving the feasibility of the proposed method. In Sections 4 and 5, discussions and conclusions are presented, respectively.

## 2. Materials and Methods

### 2.1. Control Strategy for ESS

To control voltage in a LV distribution line, a shunt-connected compensator is connected as shown in Figure 1. The distribution network is modeled as a lumped parameter model, characterized by source voltage $Us$ and short circuit impedance $Z_k$. The compensator is a 3-phase Voltage Source Inverter (VSI) equipped with any type of energy storage on the DC side. It can provide flexible active and reactive current injection to the grid and is represented by a controlled current source $I_c$. Together with the load $Z_l$, it is connected at the Point of Common Coupling (PCC).

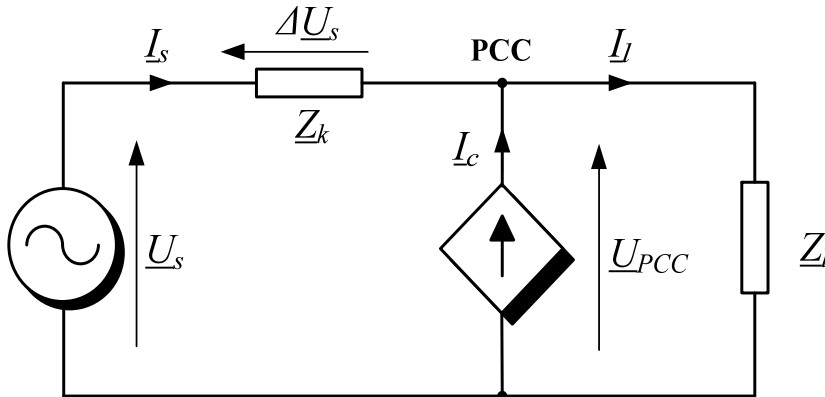

**Figure 1.** Simplified equivalent circuit of shunt active compensator installed in distribution line.

With appropriate control of injected current, the compensator can induce a voltage drop $\Delta U_s$ along the line impedance and regulate the $U_{PCC}$ voltage. Conventional shunt-voltage-regulator methods use reactive power to induce a voltage drop across the line reactance $X_k$, as shown in Figure 2a. However, in LV grids, especially in rural areas, line resistance $R_k$ is a dominant component of the line complex impedance. In this case, reactive power induces voltage drop orthogonal to the PCC voltage, which results in additional losses along the line.

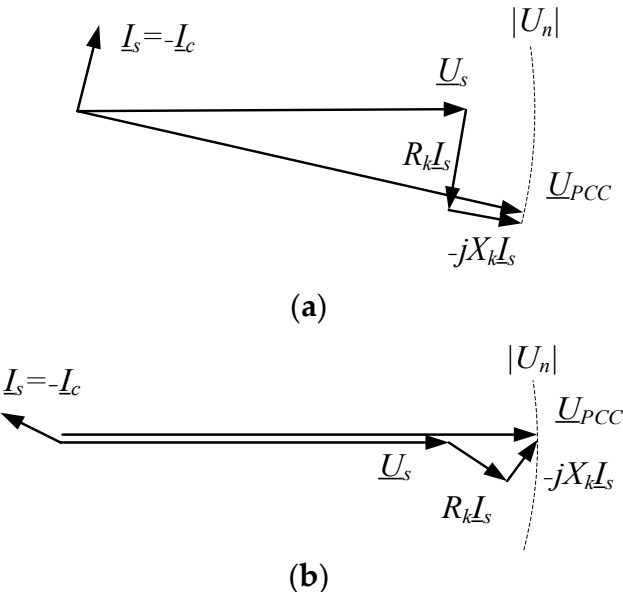

**Figure 2.** Voltage regulation: (**a**) using reactive power and (**b**) using active and reactive power with minimal current.

This drawback can be eliminated, if the compensator injects proper values of active and reactive current components. This situation is shown in Figure 2b. Here, the voltage drop induced by the injected current is parallel to the source voltage. This way, voltage regulation is achieved with minimum compensating current and, as a result, with minimum compensator apparent power, reduced line losses and no phase shift. This also helps to minimize storage capacity compared to voltage-control methods, which use only active power.

The goal of minimizing the compensating current requires information on grid-impedance parameters to calculate current set points for the compensator. Therefore, a grid-impedance-estimation method implemented in the converter control system is adopted [31]. It is a type of perturb and observe method: prior to the operation, the converter injects the fundamental components of active and reactive current and, having measured voltage response, calculates the complex grid impedance seen from the compensator terminals. It is further utilized for tuning of controllers and splitting the current into active and reactive components.

### 2.2. Controller Design

To implement the described voltage-regulation strategy, a low-level control system for the grid-interfacing power electronic converter is developed. The whole system and its control block scheme are depicted in Figure 3. It is assumed that the energy storage in the form of a battery or super-capacitor bank is connected through the DC–DC converter to the DC-link capacitor of the grid converter. It operates in closed-loop DC voltage control mode; therefore, the grid converter is free to exchange active and reactive power with the network, within its capacity.

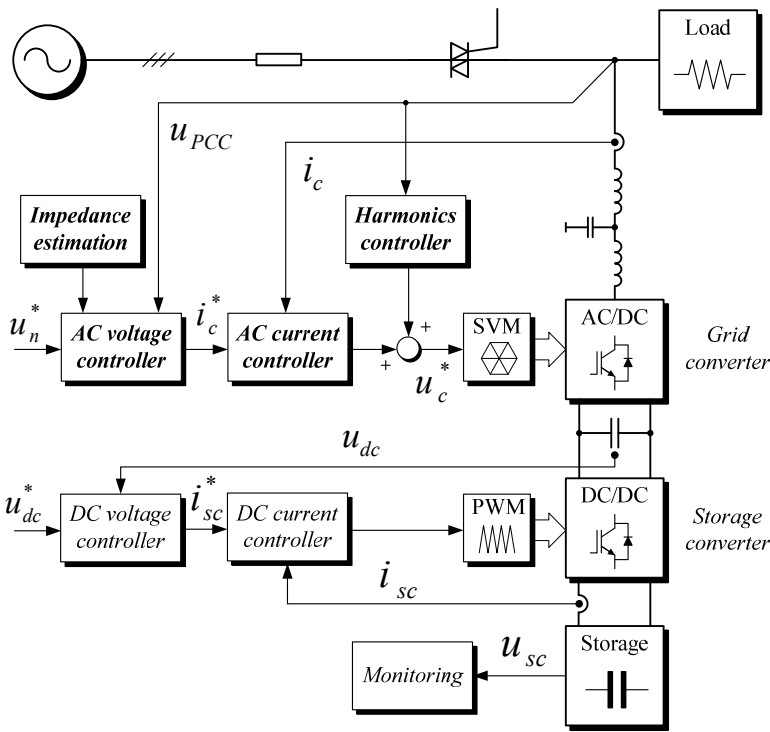

**Figure 3.** Block scheme of the proposed shunt-conditioner control system (* denotes reference values).

The control structure designed for the shunt compensator with energy storage is based on a closed-loop local grid voltage control. The control is based on the transformation of three-phase quantities to the Dual Synchronous Rotating Frame (DSRF) [32]. It is based on a two-coordinate system (*dq* frames), rotating counterclockwise with angular speed corresponding to voltage frequency. After decoupling, positive and negative voltage sequences appear as DC components in their respective frames. Finally, the DSRF algorithm estimates positive and negative voltage sequences ($u_d{}^+$, $u_d{}^-$, $u_q{}^+$, $u_q{}^-$) from $u_{PCC}$ measurements. A conventional Synchronous Reference Frame Phase-Locked Loop (SRF-PLL) synchronizes the rotating frame with the positive voltage sequence to produce the phase angle $\theta$ for all reference frame transformations used within the control structure [33]. There is no separate PLL for the negative sequence voltages, as their value is controlled to be zero. Phasor representation of voltage amplitude and unbalance is represented by phasors, as shown in Figure 4.

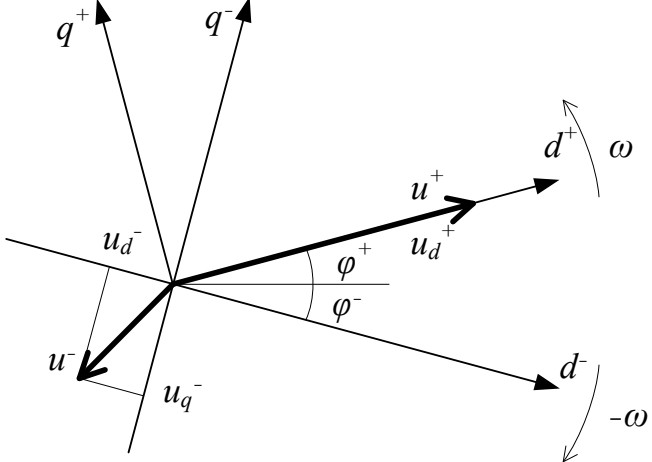

**Figure 4.** Phasor representation of voltages in DSRF.

Details of the voltage controller are shown in Figure 5. Negative voltage sequence components $u_q+$ and $u_q-$ are given as errors to the proportional–integral (PI) controllers. For the positive sequence, the difference between voltage set point $u_N$ and voltage module $u_d+$ is used as a controller error (quadrature component $u_q-$ is neglected, as it is regulated to zero by PLL). The estimated grid impedance $Z_k$ parameters are used in the voltage control loop. Its modulus is used for controller tuning, while its phase angle $\varphi_k$ allows to determine active and reactive current components, which satisfy the minimum compensator current criterion explained in Section 2.2.

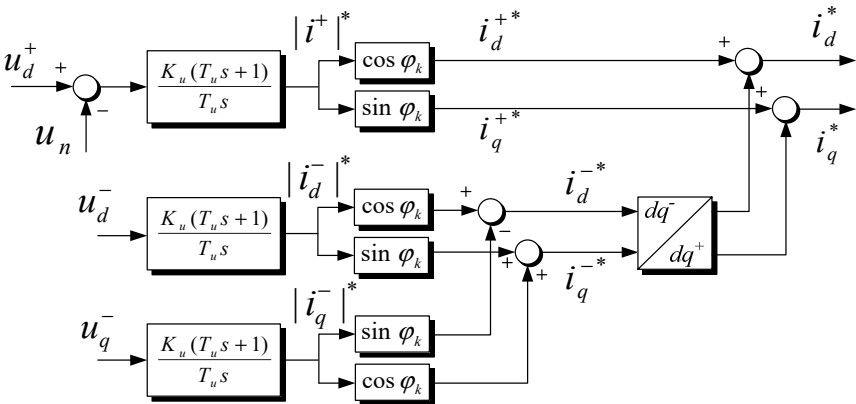

**Figure 5.** Voltage controller operating in DSRF (* denotes reference values).

Reference currents generated by the negative sequence controller are then summed up and transformed from negative to positive Synchronous Reference Frame (SRF), to create the final current reference ($i_d$*, $i_q$*) for the compensator. In this way, the positive (magnitude) and negative (unbalance) fundamental components are controlled simultaneously.

The current controller operates in the positive rotating SRF. Here, a negative sequence appears as oscillating components with double grid frequency. Typically, applied PI controllers do not provide a dynamic response high enough to control this oscillating component. To overcome this problem, a resonant path is added in parallel to the controller, as shown in Figure 6. Tuning this path to the double grid frequency results in cancellation of the steady-state error for oscillations representing negative current sequence. Gains of current PI controllers used in the algorithm are tuned using symmetrical optimum method, while gains of voltage PI controllers used in the algorithm are tuned using magnitude optimum method.

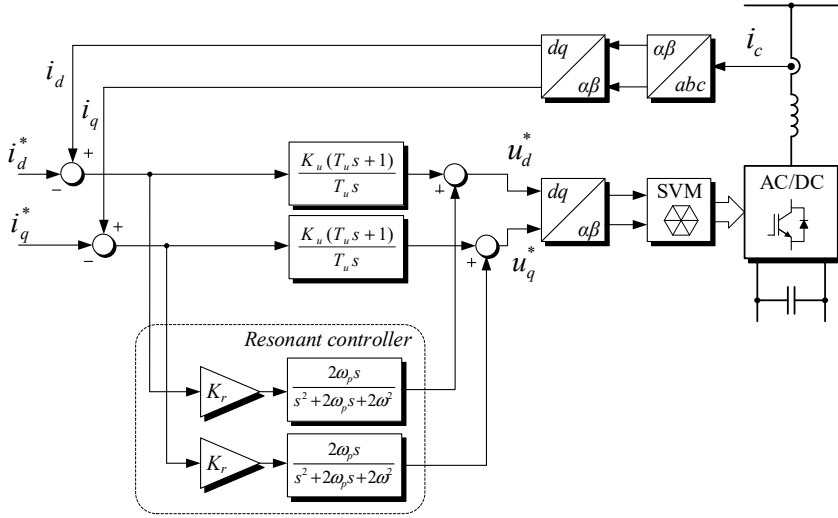

**Figure 6.** Current controller with resonant path (* denotes reference values).

Finally, Space Vector Modulation (SVM) is used to generate switching patterns for the converter transistors.

## 3. Results

The compensator control scheme was tested extensively in simulation and laboratory experiments. The model built in the SIMULINK environment included a three-level converter rated at 15 kVA with a switching frequency set at 5 kHz, connected to the $3 \times 400$V, 50 Hz grid. Selected results comparing the conventional control method and the proposed algorithm are presented below.

Figure 7 illustrates the response of the current controllers for the unbalanced grid voltage conditions (between 0.1 and 0.14 s) and unbalanced reference current (between 0.18 s and 0.22 s). A simple PI controller exhibits excellent performance in balanced conditions (Figure 7b). The step response is characterized by a single overshoot and by fast error elimination. However, during the unbalanced voltage or reference current, a double grid frequency component appears in the SRF. The PI controller dynamic is not sufficient to follow this component, and errors are relatively high.

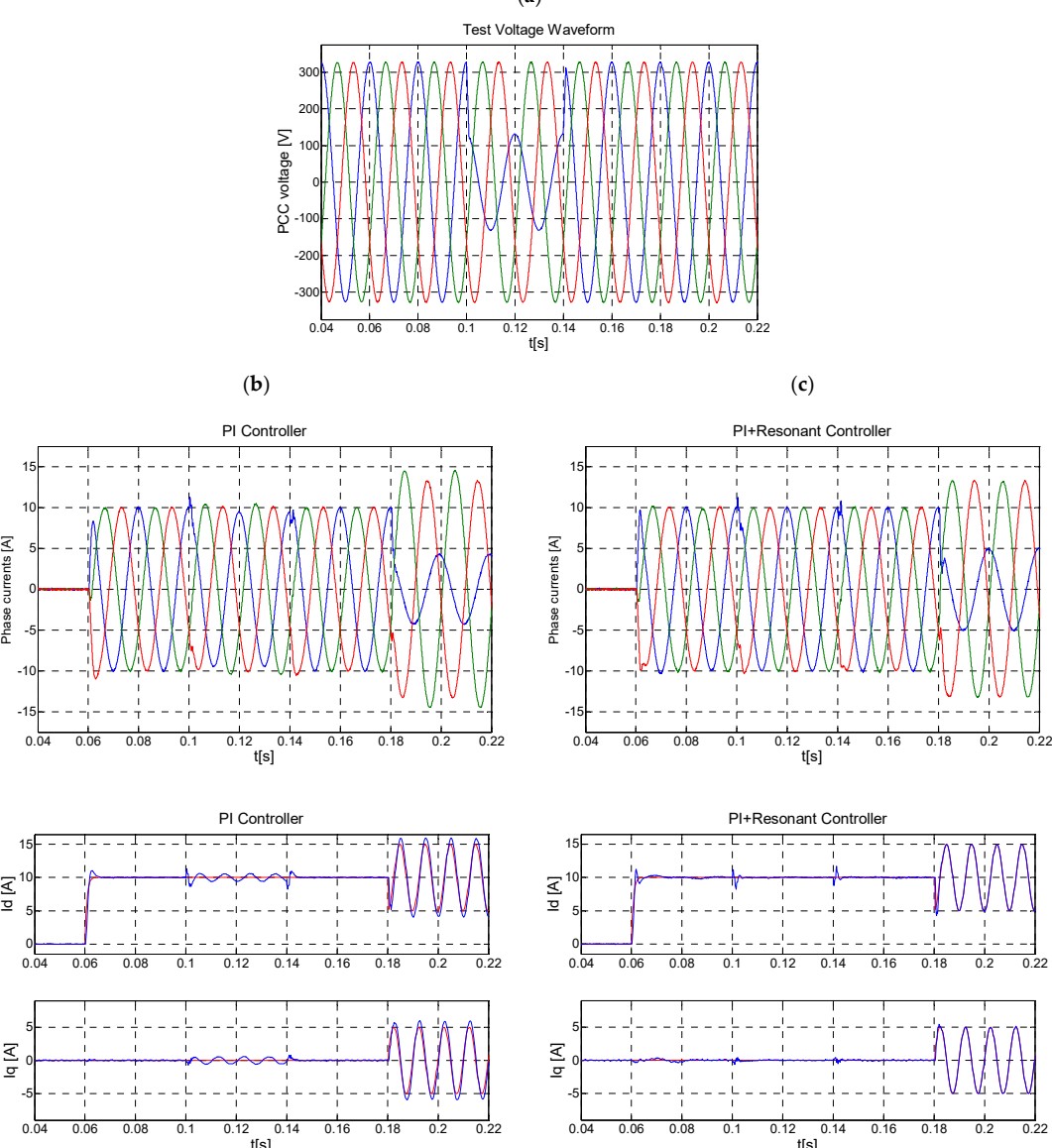

**Figure 7.** Simulation results of the current controller operating under unbalanced voltage conditions: (**a**) test voltage waveform, (**b**) PI controller and (**c**) proposed PI and resonant controller.

An additional resonant path in the current controller allows for the elimination of steady-state errors in unbalanced conditions. As shown in Figure 7c, the resonant part extends controller performance to negative sequence signals, allowing for the elimination of errors during unbalanced voltage and precisely tracked negative sequence current references. The disadvantage of such a controller is its oscillatory step response.

Figure 8 compares voltage-control strategies. A test voltage waveform includes a 10% symmetrical voltage dip between 0.4 and 0.48 s (Figure 8a). Figure 8b illustrates voltage control with a classical reactive power compensator, and Figure 8c illustrates the proposed method, which calculates active and reactive power components according to complex values of short-circuit impedance. In both cases, the presented control algorithm assures high dynamics, eliminating voltage errors during one cycle. The proposed apparent power-compensation strategy presents some advantages over the reactive power strategy. Firstly, the amplitude of the conditioner current is reduced. In such a case, the conditioner rating could be reduced, and the loading of the feeder related to voltage control is lower. Secondly, in the apparent power strategy, the conditioner action does not introduce phase distortion. In the reactive power strategy, a high variation in $u_q$ is observed, which is minimized by the PLL algorithm.

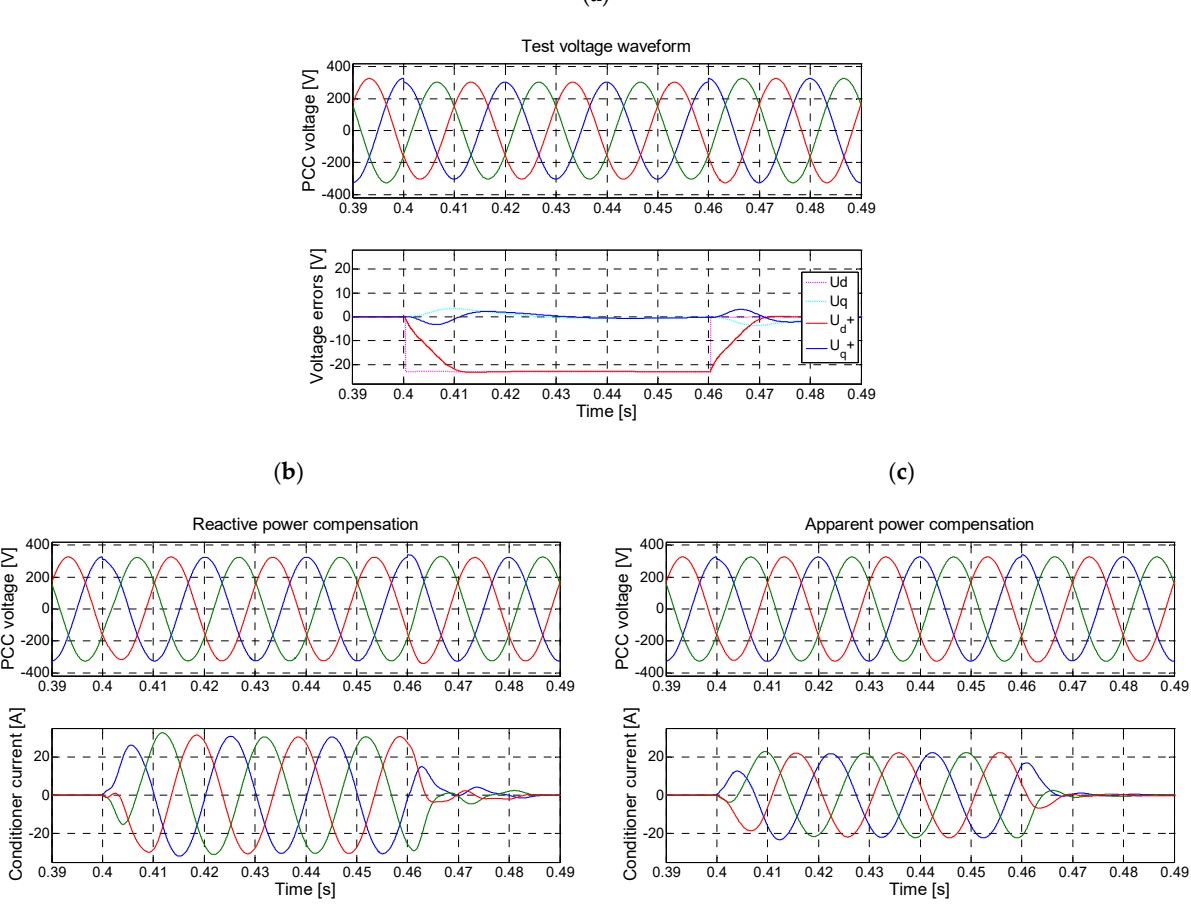

**Figure 8.** *Cont.*

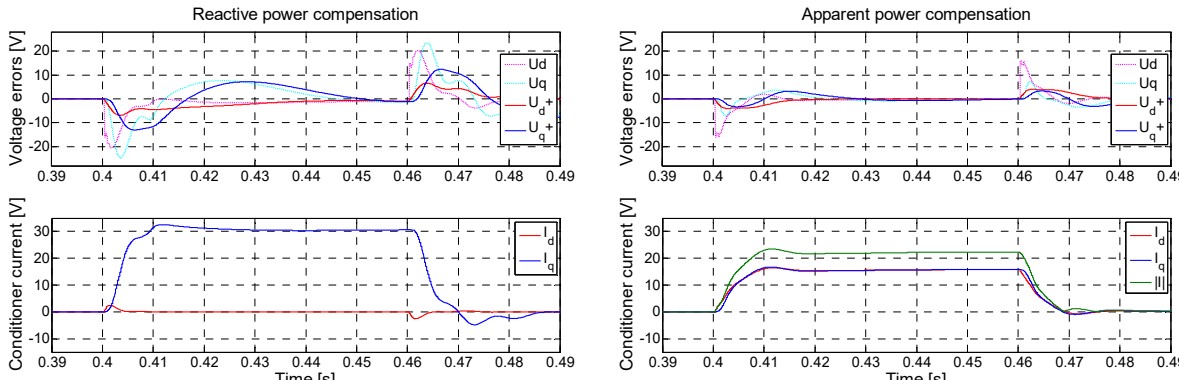

**Figure 8.** Simulation results of the voltage controller with current minimization method: (**a**) test voltage waveform, (**b**) reactive power compensation and (**c**) proposed apparent power compensation.

The power-conditioning functions of the designed system are verified experimentally. The scheme of the test setup is shown in Figure 9. The compensator is composed of a three-leg, three-level 15 kVA DC–AC grid converter and energy storage system. To provide dynamic voltage support, a power-source-type storage should be considered. In the experiment, this is a 75 V 94 F supercapacitor bank. Due to the high voltage ratio, it is interfaced to the DC-link by a dual active bridge (DAB) bi-directional DC–DC converter. The converter controls are implemented using C++ code in a dSPACE 1006 controller. The grid is simulated by a programmable voltage source with a programmable impedance module. A local resistive load bank is connected to avoid reverse power flow. A Fluke 434 power analyzer assures the measurement of the PCC voltage parameters according to the EN50160 standard. The experiments are conducted with $3 \times 120$ V nominal phase voltage (208 V line-to-line voltage). The fundamental value of the grid impedance in the experiment is: $X = 0.4 \, \Omega$, $R = 0.7 \, \Omega$ and $|Z| = 0.8 \, \Omega$.

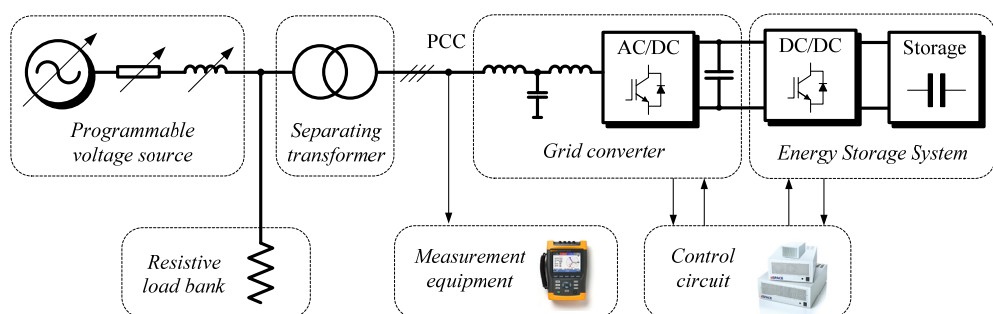

**Figure 9.** Scheme of laboratory setup.

To verify the effectiveness of the PCC voltage control, a steady-state voltage regulation test was conducted. The conditioner regulates voltage to the nominal value using two control strategies: reactive power injection and apparent power injection with minimum current. The resulting measurement data are shown in Figure 10. In Figure 10a, the voltage at the PCC with nominal (left) and reduced (right) amplitude, without compensator, is shown. Figure 10b shows the result of voltage regulation by the conditioner using reactive power only. Using 4.5 kVA reactive power, the voltage amplitude is compensated to match the nominal amplitude. Figure 10c illustrates the same scenario of voltage regulation, though with the use of apparent power. The active and reactive components are set with the same properties for grid resistance and reactance. It can be observed that the apparent power used for compensation is significantly reduced from 4.5 kVA to 2.5 kVA.

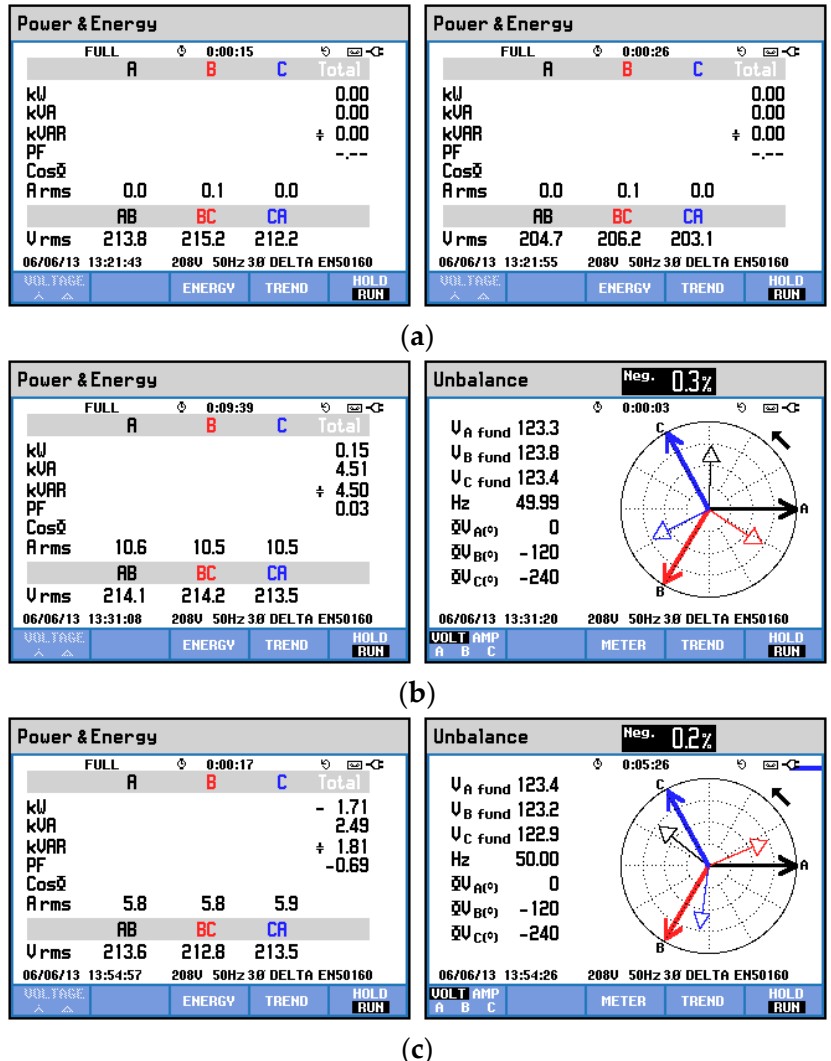

**Figure 10.** Voltage regulation in steady state: (**a**) nominal and reduced voltage without compensation, (**b**) reactive power compensation and (**c**) apparent power compensation.

In Figure 11, the steady-state performance of the voltage balancing is presented. To balance the voltages, the conditioner injects a negative sequence current with active and reactive components set according to the grid complex impedance. The voltage drop on the grid impedance results in voltage balancing at the PCC. As seen in the graphs, the voltage unbalance can be eliminated. The integral path of the voltage controllers eliminates the steady-state error.

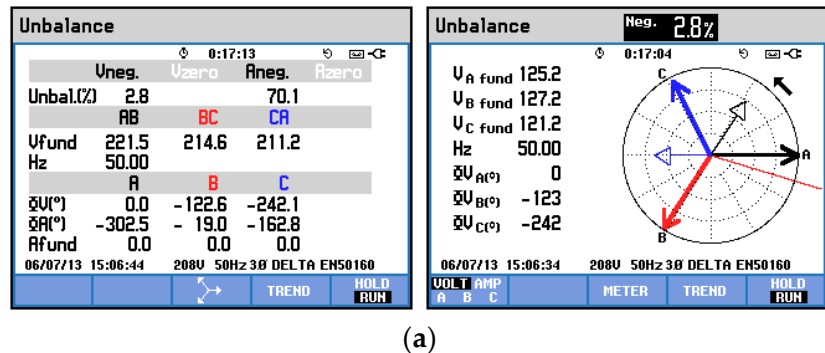

**Figure 11.** *Cont.*

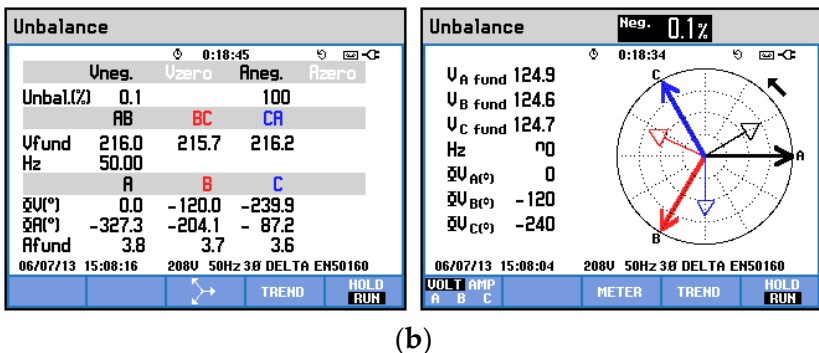

**Figure 11.** Three-phase voltage unbalance: (**a**) without compensation and (**b**) with compensation.

Figure 12 shows the compensation of an unbalanced voltage dip: 70% single phase dip with 100 ms duration. The conditioner operates in minimum apparent power control mode and with a three-phase voltage-balancing controller. The grid impedance is increased in the experiment to $X = 0.65\ \Omega$, $R = 1.7\ \Omega$ and $|Z| = 1.8\ \Omega$ in order to extend the conditioner control range and improve the visibility of the results.

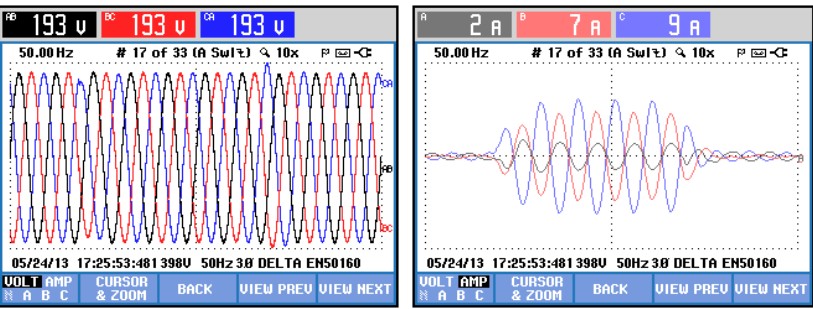

**Figure 12.** Single-phase 70% dip compensation.

The compensating action brings the voltage amplitude to the nominal level in less than one cycle. In this case, the conditioner current is unbalanced, as a response to the negative sequence in the grid voltage. The characteristic feature of the PI-based voltage control is a transient overvoltage at the end of the voltage dip. This is caused by the integral action of the controller, which maintains the voltage-boost action for an instant after the voltage returns to a nominal level.

## 4. Discussion

Compared to classical control structures that use only active or reactive power for grid voltage control, the proposed control scheme includes:

- Utilization of active and reactive power for voltage regulation;
- Minimization of the conditioner current amplitude;
- Estimation of the grid impedance;
- Voltage balancing.

The injection of active and reactive components proportional to the R/X grid impedance parameters helps to minimize the storage capacity and minimize line current/losses compared to voltage regulation using only active or reactive power. The proposed method is, therefore, superior to others that only utilize storage active power control or reactive power compensation.

Universal power quality improvement is performed in terms of voltage magnitude and unbalance as well as in static and dynamic states. The low-level converter control results in very high dynamics of voltage regulation and the mitigation of temporary events. As indicated in Figure 3, an additional harmonic controller can be implemented in the

proposed solution, in parallel to the fundamental component. It was proven that voltage harmonic reduction is possible using control methods similar to the ones described in [34].

The most proper implementation of this method is by using ESS or V2G interfaces in rural areas, i.e., for compensation of voltage fluctuations caused by PV and heavy loads. The proposed method can be also adopted to DER-interfacing converters. Their only flexibility is reactive power and unbalance control, and, hence, minimization of the compensating current would not be possible. Moreover, to compensate unbalance, the DC-link capacitor needs to be oversized due to the double-grid-frequency power oscillations.

A short-term balancing is shown in the given examples; however, with an extended storage capacity, long-term balancing is also possible. A feasible option would be to minimize the daily voltage fluctuations related to the load profiles or PV generation—performing peak shaving. Practically, this would mean storing energy during high production/low consumption hours and releasing energy during low production/high consumption hours. In case of high PV penetration, this would exhibit daily profiles similar to those reported in [17]. This has, therefore, the advantage of minimizing feeder loading and increasing hosting capacity of DERs.

To minimize the storage capacity, power rating and energy circulation in the system, an adaptive reference voltage can be used, as shown, e.g., in [30]. The reference voltage can be gradually varied between a minimum and a maximum value, as permissible by the standards, which will prevent violations at a minimal implementation cost. If more compensating devices would be connected along one feeder, a coordinated droop method or voltage dead zones should be applied to allow for power sharing. The abovementioned issues related to the parallel operation of multiple storage devices are considered as part of the authors' future work.

## 5. Conclusions

This article discusses the analysis and development of a control method for energy storage systems connected in shunt to the network grid and used for voltage-quality and hosting-capacity improvement in LV distribution networks. By adjusting the active and reactive current components according to the estimated grid impedance, the designed controllers minimize the current required to perform voltage regulation compared to methods utilizing only active or reactive power. With instantaneous symmetrical components' decomposition, voltage balancing is also performed with a performance superior to simple control methods based on instantaneous voltage magnitude.

As confirmed in the experiment, the proposed method is effective for dynamic grid voltage regulation. It helps to reduce rapid voltage changes (RVC), flicker and voltage dips and rises, and it compensates for voltage unbalance. With extended storage capacity, this method is useful for long-term voltage regulation and may reduce the over/undervoltages caused by excessive line loading or generation.

The proposed ESS controller works independently of PV inverters and loads and can be integrated into existing grids by applying it to any energy storage system. It is based on local voltage control, so there is no need to invest in a costly ICT infrastructure. Automatic grid impedance estimation makes the proposed method a 'plug and play' solution, as controllers can be tuned automatically. Future work includes analyses of the optimal placement and sizing of compensator parameters in LV networks, aiming at the deployment of such compensators in real distribution networks.

**Author Contributions:** Methodology, validation, funding acquisition, K.R.; investigation, writing—review and editing, J.B.; conceptualization, formal analysis, S.B.; supervision, resources, P.C. All authors have read and agreed to the published version of the manuscript.

**Funding:** The work was financially supported by the National Centre for Research and Development via Grant No. LIDER/30/0166/L-10/18/NCBR/2019.

**Institutional Review Board Statement:** Not applicable.

**Informed Consent Statement:** Not applicable.

**Data Availability Statement:** Not applicable.

**Conflicts of Interest:** The authors declare no conflict of interest.

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
