# Peer review of "Dynamic Voltage Regulation and Unbalance Compensation in a Low-Voltage Distribution Network Using Energy Storage System"

_applsci, doi:10.3390/app122211678_

Round 1
Reviewer 1 Report
1. It’s suggested to improve abstract by including more details about existing challenges and the proposed solution.
2. More details about Voltage Regulation and Unbalance Compensation in a Distribution Network Can be included from SPSO Based Optimal Integration of DGs in Local Distribution Systems under Extreme Load Growth for Smart Cities and A resonant damping control and analysis for LCL-type grid-connected inverter.
3. Authors should emphasize about their contributions.
4. How authors claim that their proposed controller design is better? Its suggested to give comparison with the state of the art approaches.
5. Conclusions should be improved with future directions.
Author Response
Dear Reviewer,
Thank You for reviewing the manuscript and please find responses to Your specific suggestions below.
- It’s suggested to improve abstract by including more details about existing challenges and the proposed solution.
Abstract has been updated and extended. More information is given in the paper introduction.
- More details about Voltage Regulation and Unbalance Compensation in a Distribution Network Can be included from SPSO Based Optimal Integration of DGs in Local Distribution Systems under Extreme Load Growth for Smart Cities and A resonant damping control and analysis for LCL-type grid-connected inverter.
Thank You for suggestions. First paper presents a viable alternative for voltage regulation, so it has been included to the literature overview. The other one focuses on a specific problem of the filter design and resonance damping, which is important for the converter design, but is not the main focus of the paper.
- Authors should emphasize about their contributions.
Original contribution of the research has been stated in the abstract and at the end of the introduction. Specific authors contributions are provided in the dedicated paragraph at the end of the article (as required in the instructions for authors).
- How authors claim that their proposed controller design is better? Its suggested to give comparison with the state of the art approaches.
Additional simulation results are added to the paper to illustrate advantages of the proposed controller over the conventional approach. The authors have shown this in the added Figures 7 and 8.
- Conclusions should be improved with future directions.
Conclusions are extended and future work is added in the revised manuscript.
Reviewer 2 Report
The following are my comments.
1. The title of the paper looks vague. Please change it.
2. The novelty of the paper must be clearly defined in the abstract section.
3. Literature study needs further improvement.
4. The authors are suggested to put the list of contributions at the end of the introduction.
5. There is no mathematical proof of the controller proposed in the paper. It is strongly recommended you to prove the controller behavior mathematically.
6. How are the PID gains obtained?
7. What are the impacts of different PID gains?
8. HIL information is not clear in the paper.
9. It will be good if you compare the HIL results with the simulated output.
10. How can you prove the proposed controller is good?
11. There is no comparison between the existing controllers and the proposed controller
12. Include the future scope of the paper in the conclusion section.
13. Finally, check for the grammatical mistakes and typos.
Author Response
Dear Reviewer,
Thank You for revising the manuscript. Please find responses to Your specific suggestions below.
- The title of the paper looks vague. Please change it.
New title is proposed: “Dynamic Voltage Regulation and Unbalance Compensation in a Low Voltage Distribution Network using Energy Storage System”
- The novelty of the paper must be clearly defined in the abstract section.
Abstract and introduction sections have been updated to clearly state novelty of the paper and original authors contribution.
- Literature study needs further improvement.
Literature study has been updated and references formatting has been corrected.
- The authors are suggested to put the list of contributions at the end of the introduction.
Original contribution of the research has been stated in the abstract and at the end of the introduction. Specific authors contributions are provided in the dedicated paragraph at the end of the article (as required in the instructions for authors).
- There is no mathematical proof of the controller proposed in the paper. It is strongly recommended you to prove the controller behavior mathematically.
Authors did not prove of the controller stability analytically. Well known PI controllers were applied and their gains were tuned according to widely accepted method (moduls optimum). Stability of the controllers was verified by simulations and experiments.
- How are the PID gains obtained?
Current PI controllers are tuned using symmetrical optimum method to obtain fast and stable response in dynamic states.
Voltage PI controllers are tuned using magnitude optimum method to eliminate controller overshoot that would result in voltage distortions while providing stable and relatively fast response.
- What are the impacts of different PID gains?
Different gains were tested during simulation studies. Lower gains led to decrease in control dynamics, while higher gains resulted instability.
- HIL information is not clear in the paper.
There is no HIL information in the paper, this method was not used during the research. Testing of the solution was done using a laboratory setup including real 3-level voltage source inverter and a programmable DSP platform with use of programmable AC source.
- It will be good if you compare the HIL results with the simulated output.
Prior to implementation in the laboratory setup, the algorithm was extensively tested using Simulink model. Additional simulation results were added to the paper to clarify the controller response.
- How can you prove the proposed controller is good?
Simulation results show superior performance in comparison to conventional controllers. Experimental results show significant improvement in power quality indices during operation of the compensator with proposed control algorithm.
- There is no comparison between the existing controllers and the proposed controller
Additional simulation results are added to the paper to illustrate advantages of the proposed controller. Performance of the controller and compensator current magnitudes are compared.
- Include the future scope of the paper in the conclusion section.
Future work is added at the end of conclusions in the revised manuscript.
- Finally, check for the grammatical mistakes and typos.
Paper has been checked and corrected.
Round 2
Reviewer 1 Report
I have no further comments
Reviewer 2 Report
I am satisfied with the responses. The paper may be forwarded for further action.